# Neural Distribution Learning for Generalized Time-to-Event Prediction

## Abstract

Predicting the time to the next event is an important task in various domains. However, due to censoring and irregularly sampled sequences, time-to-event prediction has resulted in limited success only for particular tasks, architectures and data. Using recent advances in probabilistic programming and density networks, we make the case for a generalized parametric survival approach, sequentially predicting a distribution over the time to the next event. Unlike previous work, the proposed method can use asynchronously sampled features for censored, discrete, and multivariate data. Furthermore, it achieves good performance and near perfect calibration for probabilistic predictions without using rigid network-architectures, multitask approaches, complex learning schemes or non-trivial adaptations of cox-models. We firmly establish that this can be achieved in the standard neural network framework by simply switching out the output layer and loss function.

## 1 Introduction

Many real-world tasks can be formulated as time-to-event (TTE) prediction problems, forecasting the time taken until an event-of-interest happens in the future. Examples are predicting time to the onset of a particular disease (Price et al., 2017), predicting when a machine fails (Salfner et al., 2010) or predicting future user logins (Sobaszek & Gola, 2016). Unfortunately, the data often consists of temporal features and recurrent events which are sparsely and irregularly sampled over time. This makes the data challenging to work with. Furthermore, the main challenge is that if we do not observe a terminating event for a sequence, the time to event becomes *right censored* (Klein & Moeschberger, 2005). This essentially means that we only know that the event will take place later than when it was known to *not* occur.

Proposed solutions taking account for censoring includes predicting TTE binned across fixed time windows (classification approaches (Harutyunyan et al., 2017; Lee et al., 2018)), predicting TTE point-wise (regressive approaches (Ishwaran et al., 2008)), ranking the risk of subjects (semi-parametric (Katzman et al., 2016; Luck et al., 2017) and non-parametric (Kalderstam, 2015; Chen et al., 2013)), or estimating the target and the feature distributions jointly (variational, stochastic formulations (Soleimani et al., 2017; Ranganath et al., 2016; Mei & Eisner, 2017; Xu et al., 2017)). Numerous works have used combinations of these approaches (Du et al., 2016; Li, 2017; Xiao et al., 2017b;a; Harutyunyan et al., 2017; Lee et al., 2018). However, most existing solutions have problems with tight couplings to particular types of queries, restrictive and data-dependent loss formulations, complex model architectures, or improper handling of censored data.

While there has been no lack of novelty in the proposed solutions, we find that the seemingly most straightforward generalized parametric survival approach has lacked a thorough treatment. In this paper, we formulate the TTE prediction task as, in each timestep, predicting all parameters of some discretized probability distribution over the current time to the next event. We show the benefits of defining distributions in terms of *cumulative hazard functions* (Sec. 3.1). Finally, we show that the regular negative log-likelihood for right-censored data, *censored log-likelihood* (known at least since de Moivre (1731)) as the loss function is sufficient to yield near perfect calibration. All together, this leads to an obvious yet novel framework for general-purpose TTE prediction. We will refer to this as *HazardNet*. While generalizing previous work, it is easy to implement and evaluate while being capable of yielding real-time predictions on various inference queries, such as predicting the probability of a user returning in 30 days, the expected time to failure or median time of survival.

In section 4 we show that the model makes unbiased, calibrated probabilistic predictions and efficiently utilizes training data. This is shown by evaluating on three publicly available datasets comparing multiple TTE-distributions and neural network architectures. For the purpose of establishing a strong baseline, we propose a novel evaluation scheme, designed for real-life applications. In addition, in section 5 we reformulate the binary task of musical onset detection into a multivariate distribution prediction task, achieving state of the art results.

## 2  RELATED WORK

There is a diverse array of problem formulations for TTE-prediction (Wang et al., 2017). The methods for censored data are mainly based on classical semi-parametric Cox-models for continuous target values (Katzman et al., 2016; Luck et al., 2017; Joshi & Reeves, 2006). Others formulate it as classification problems (Harutyunyan et al., 2017; Lee et al., 2018) or multi-task learning (Luck et al., 2017; Lee et al., 2018) by predicting the event probability in each step for a fixed number of steps $\tau$ ahead together with a ranking loss. Non-parametric learning-to-rank methods have been proposed (Kalderstam, 2015) but they suffer from scalability problems. Some studies (Du et al., 2016; Li, 2017; Xiao et al., 2017b;a) predict both TTE and classification of next event jointly using composition of loss functions. Others have learned from censored data by predicting features and target jointly, either using Gaussian processes (Soleimani et al., 2017) or deep exponential families (Ranganath et al., 2016). Recent extensions of this work (Miscouridou et al.) includes an interesting discussion on methods for approximating arbitrary distributions, discretization, evaluation and missing feature data. Sequential prediction is often formulated as asynchronously predicting at the same time as the event (Du et al., 2016; Li, 2017; Xiao et al., 2017b;a; Lee et al., 2018; Avati et al., 2018). However, this makes it unclear how to make predictions between events. For features arriving at different times, it has been proposed to use two recurrent neural networks (RNNs) acting on different timescales (Xiao et al., 2017b), but it did not deal with censored data. For the general problem of unevenly spaced sequences, there has been other notable successful neural network approaches such as Phased LSTM (Neil et al., 2016) or Time-LSTM (Zhu et al.).

**Limitations:**  Most solutions are designed specifically for each task, such as rank prediction (Katzman et al., 2016; Luck et al., 2017; Kalderstam, 2015; Chen et al., 2013), classification (Harutyunyan et al., 2017; Lee et al., 2018), and more. When using stochastic formulations (Soleimani et al., 2017; Ranganath et al., 2016; Mei & Eisner, 2017; Xu et al., 2017), inference requires extra steps and are arguably made for other tasks (such as missingness of data or understanding feature importance). Without taking into account censored data (Du et al., 2016; Li, 2017; Xiao et al., 2017b;a) probabilistic predictions might have little meaning. Most models are based on strong assumptions about data or underlying distributions, and are difficult to adapt to new problems. The pure parametric survival approaches based on density networks either focused on specific distributions (Martinsson, 2016; Avati et al., 2018) or made cases against the use of regular log-likelihood (Avati et al., 2018; Ren et al., 2018). While there has been recent interest in the importance of calibration (Luck et al., 2017; Avati et al., 2018), we argue for a different method of evaluating it in a real-world setting, and most importantly we want to establish a baseline to compare against whether the regular log-likelihood loss needs improvement at all.

**Our contribution:**  Building on the work of Martinsson (2016), we present a flexible and generalizable framework for time to event prediction that can work with multiple model architectures and distributions, supporting various probabilistic inference queries and handle asynchronously arriving features. We found no in-depth experiments or convincing results on how to evaluate performance, especially calibration for probabilistic predictions. To this end, we propose an evaluation strategy in Section 4. We found no general discussions on learning parametric survival distributions in this setting nor any in depth examples using the *Weibull*, *Pareto*, *LogLogistic*, their mixtures, or other distributions as discretized forms. By thoroughly investigating what ought to be a standard approach and suggesting evaluation methods for it, we hope to establish a strong baseline for time to event prediction. The implementation will be released online.

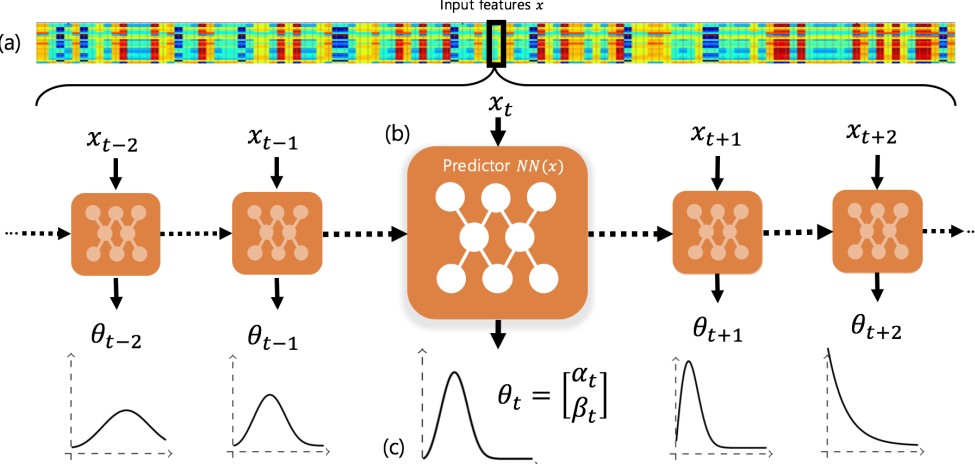

Figure 1: Schematic illustration of HazardNet. (a) We feed a sequence of feature vectors $x$, shown as a heatmap, to an arbitrary model $NN$ (i.e., a *predictor*). (b) $NN$ predicts the parameter of a distribution, $\theta_t = NN(x_t)$, for each step $t$ as an output (c). As an example, we can use an RNN to sequentially predict Weibull distribution parameters $\theta_t = [\alpha_t, \beta_t] = NN(x_{0:t})$.

## 3 HAZARDNET

While many previous approaches have focused on specific data distributions or tasks, our goal is to provide a general framework by focusing on the problem formulation for TTE-prediction itself. We frame it as a Distribution Learning problem.

The basic idea is a density network coupled with a parametric survival approach. Let a neural network predict parameters of some TTE-distribution and train it using log-likelihood for censored data. As shown in Figure 1, in every step $t$ we let the *predictor* map features $x_t$ to predicted parameters $\theta_t$, governing the shape of a distribution. The distribution is specified by fixing a functional form of a cumulative hazard function $\Lambda$. In the following sections we will show how to use discrete target values and the possibility of learning to approximate arbitrarily complex distributions.

### 3.1 EFFICIENTLY WORKING WITH DISTRIBUTIONS USING CUMULATIVE HAZARD FUNCTIONS

We will employ the notational convenience of using **cumulative hazard functions** (CHF) to define probability distributions. In the survival analysis context, it is common to focus on *hazard functions*, but we found it more efficient to focus on its integral, defined in the following.

**Definition 1.** A *cumulative hazard function* $\Lambda : \mathbb{R}_+ \to \mathbb{R}_+$ is a monotonically increasing positive function such that for all $x \geq 0$, $\epsilon \geq 0$,

$$0 = \Lambda(0) \leq \Lambda(x) \leq \Lambda(x + \epsilon) \leq \Lambda(\infty) = \infty \tag{1}$$

where the *hazard* function $\lambda(x) = \frac{\partial}{\partial x}\Lambda(x)$ is its positive derivative if it exists.

A cumulative hazard function $\Lambda(x)$ is a straightforward way of representing a positive distribution. If $X$ is a random variable, we can write its corresponding *cumulative density function* as $F(x) = Pr(X < x) = 1 - e^{-\Lambda(x)}$ or conversely, $\Lambda \equiv -\log(1 - F)$. Its *probability density function* can be written $f(x) = \frac{\partial}{\partial x}F(x) = \lambda(x)e^{-\Lambda(x)}$. If we let the random variable $X$ represent a life span, the *survival function* $S(x) = \Pr(X > x) = 1 - F(x) = e^{-\Lambda(x)}$ is the probability of surviving until time $x$.

Cumulative Hazard functions makes a good abstraction for implementation since the conditions of Equation 1 are easily verifiable. It also simplifies transition from continuous to discrete distributions, an important consideration since real-world data tend to be discrete. Given $\Lambda(x)$, we can easily form a discrete distribution by defining a *probability mass function* $p(t) = S(t) - S(t + 1) =$

Table 1: Examples of cumulative hazard function representations of distributions.

| $\Lambda_{Exponential}$ | $\Lambda_{Weibull}$ | $\Lambda_{Pareto}$ | $\Lambda_{LogLogistic}$ | $\Lambda_{LogLogisticMixedHazards}$ |
|---|---|---|---|---|
| $\left(\frac{x}{\alpha}\right)$ | $\left(\frac{x}{\alpha}\right)^{\beta}$ | $\beta \log\left(1 + \frac{x}{\alpha}\right)$ | $\log\left(1 + \left(\frac{x}{\alpha}\right)^{\beta}\right)$ | $\sum_{k} \Lambda_{LogLogistic,k}$ |

$e^{-\Lambda(t)} - e^{-\Lambda(t+1)}$ for $t = 1, 2, \ldots, \infty$. This also makes it easy to use different levels of discretization (i.e., days to weeks). In doing so, instead of using a continuous loss function after discretization, we can learn the discretized distribution directly, which in turn can be used to approximate a continuous distribution if needed.

**Example** The exponential distribution $f(x) = \frac{1}{\alpha}e^{-\frac{x}{\alpha}}$ can be defined with the CHF $\Lambda_{Exponential}(x) = \frac{x}{\alpha}$. Discretization yields the geometric distribution, with probability mass function $p(x) = e^{-\frac{x}{\alpha}} - e^{-\frac{x+1}{\alpha}}$.

### 3.2 COMPOSING MIXTURES OF DISTRIBUTIONS

Another clear benefit of the Cumulative Hazards-perspective of distributions is that it simplifies the creation of new distributions. Recent developments on simplifying probabilistic programming have made it possible to effectively compose and work with distributions and their mixtures (Siddharth et al., 2017; Dillon et al., 2017; Tran et al., 2016). One of the most well-known form of distribution composition is the mixture-density network (MDN) (Bishop, 1994), where the probability density $f = \sum_{k} w_k f_k$ of the target data is a linear combination of more simple basis distributions. From the perspective of CHFs, we propose that it is simple to create other powerful compositions that can easily be learned from censored data. It is easy to show that the space of CHFs are closed under strictly increasing mappings ($\Lambda_a^2$), composition ($\Lambda_a(\Lambda_b)$), multiplication ($\Lambda_a \cdot \Lambda_b$), addition $\Lambda_a + \Lambda_b$, and multiplication with positive scalars. As a simple example, $\Lambda_{LogLogistic} = \Lambda_{Pareto}(\Lambda_{Weibull})$. Similar to MDNs, we can create expressive possibly multimodal distributions by composing simpler CHFs, which can be easily learned with censored data and be discretized to any time resolution. In section 4, apart from using the commonly used Weibull-, Pareto[1]- and LogLogistic distributions we try their more expressive additive compositions respectively (named *MixedHazards*). We do this in order to show the generality of our approach and how it works with arbitrary distributions.

### 3.3 CENSORED LOG-LIKELIHOOD

Under mild assumptions discussed here, we can accurately do maximum likelihood estimation using censored data utilizing *censored log-likelihood*. With $Y$ a random variable of interest parametrized by $\theta$, c a constant or random variable of censoring time s.t $Y \perp c|\theta$, $X = \min(Y, c)$ the censored (truncated) random variable of interest, and $U = [Y \leq c]$ is the non-censoring indicator, (with $u = 1$ indicating *not censored*). Under these conditions it is well known that the likelihood of a censored (Klein & Moeschberger, 2005) random variable $X = x$ with an observed non-censoring indicator $U = u$ can be factored as

$$L(X = x, U = u) \propto f(x)^u S(x)^{1-u} = \begin{cases} f(x|\theta) & x < c & \text{(uncensored)} \\ Pr(Y > c|\theta) & x = c & \text{(right censored)} \\ 0 & x > c & \text{(impossible query)} \end{cases}$$

Which also holds in discretized case, where the above are discretized as $(Y_d, c_d, X_d) = (\lfloor Y \rfloor, \lfloor c \rfloor, \lfloor X \rfloor)$ and $U_d = [Y_d < c_d]$ is the non-censoring indicator (Martinsson, 2016). The practical implications of this assumption is that $c$ should not be predictable by the features. Our experiments (Sec. 4) show that we learn proper calibration, hence this assumption can be made to hold in practice.

The log-likelihood for continuous (Eq. 2) and discrete (Eq. 3) observations $(x, u)$, can be simplified as below.

---

[1]Parametrized as the Lomax distribution.

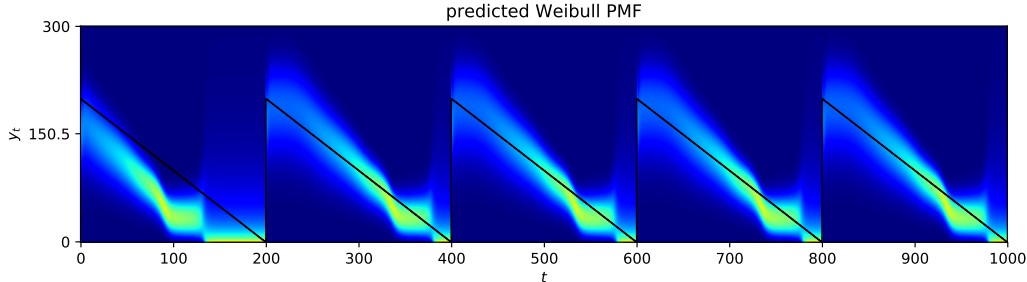

Figure 2: Predicted Weibull pmf as a heatmap, HazardNet prediction with single 2-node LSTM cell fitted on evenly spaced events, using lagged event indicator as only input. Marked line is the actual target TTE, a countdown reaching $Y_t = 0$ at the time of event.

$$\mathfrak{L}(\theta, x, c) = \log \left[ f(x)^u S(x)^{1-u} \right] = u \cdot \log \left[ \lambda(x) \right] - \Lambda(x) \tag{2}$$

$$\mathfrak{L}_d(\theta, x, c) = \log \left[ p(x)^u S(x+1)^{1-u} \right] = u \cdot \log \left[ e^{\Lambda(x+1) - \Lambda(x)} - 1 \right] - \Lambda(x+1) \tag{3}$$

In this work we focus on the discrete log-likelihood as loss function, using the CHFs in Table 1.

### 3.4 LOSS FUNCTION FOR SEQUENTIAL PREDICTION

Consider a sequence indexed by $t = 1, \dots T$. We model the possibly censored observation $(x_t, c_t)$ of time to event and censoring time at timestep $t$ as a realizations of some random variable $Y_t$, censored using a known censoring time $c_t$ s.t $X_t = \min(Y_t, c_t)$. The optimization task is to predict parameters $\theta_t$ of a target distribution, given by a fixed form of $\Lambda$, which maximizes the log-likelihood of the observation. The proposed (discrete) loss for one sequence is thus $\sum_t^T -\mathfrak{L}_d(\theta_t, x_t, c_t)$.

While this might seem like an obvious formulation, most prior work has chosen different paths. Most work has focused on specifics of certain continuous distributions. While widely known, we hope to reaffirm that the principles of parametric survival analysis extends to all positive and discrete distributions. In terms of data shape, the dominant theme is to asynchronously predicting continuous inter-arrival times (Du et al. (2016); Li (2017); Xiao et al. (2017b;a); Lee et al. (2018); Avati et al. (2018) etc) assuming the time of prediction to coincide with the event times. We focus on the discrete form of TTE, predicted from arbitrary but equally-spaced timesteps.

Equal-spacing and discrete TTE addresses two problems. First, to use sequence models we need a fixed order and want to reduce the maximum sequence length. With continuous time, the number of events can be high, and the order of events may be unclear due to tied event times. Both of these issues are solved through binning and discretization. A second problem is that with unevenly-spaced predictions the loss function involves unevenly spaced terms over time. The adequacy of this scheme when using arbitrary distributions is unclear. A subsequent question becomes how to predict or incorporate features arriving between events. By predicting at fixed intervals these are both solved since we decoupled the prediction and the event times. We have found no work addressing all these questions for censored time to event. Apparent drawback with our method is that if we want to train on a duration of 1000 time-units of data it yields a possibly sparse sequence of length 1000 (as in Figure 2). It also limits the frequency of inference to the hop size (i.e. daily predictions).

## 4 EXPERIMENTAL RESULTS

We compare a diverse number of time to event distributions (Table 1) learned with different network architectures (Figure 3) on three different datasets (Table. 2) against a binary baseline.

### 4.1 DATA DESCRIPTION

We used three publicly available event-log datasets close to real-world applications. LastFM-1k (Celma, 2010) contains complete listening histories of about 1000 users. BPI (Dees & Van Dongen, 2016) includes one year of click stream data from 26k users who logged in an Employee Insurance

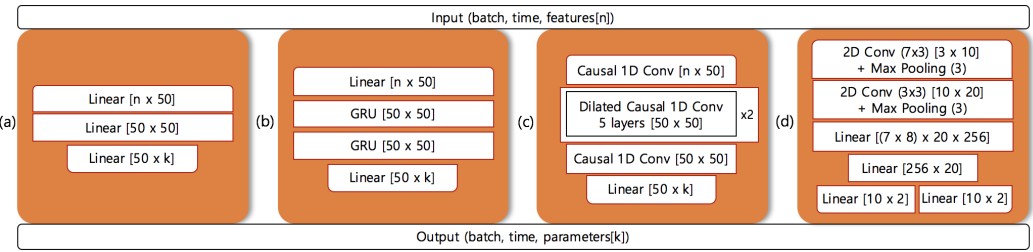

Figure 3: Models used in experiments with details. $n$ is the number of features, $k$ is the number of distribution parameters. (a) MLP (b) RNN (c) Dilated Causal CNN (d) Specific CNN architecture for musical onset detection discussed in Section 5.

Table 2: Dataset details

| dataset | sequences | length(days) | features | sequence id | event |
|---|---|---|---|---|---|
| LastFM-1k | 991 | 1538 | 2 | user id | 'song played' |
| BPI Challenge 2016 | 26613 | 242 | 8 | user id | 'click event' |
| Linux git logs | 13432 | 2664 | 5 | contributor | 'commit' |

Agency office website. We created the Linux commit log dataset, which contains all commits from 13k active contributors since 2011. [2]

For each dataset, we discretized the resolution from millisecond to daily measurements. In doing so, a day with multiple events is considered a day with event hence its discrete time to event is zero. Features were aggregated by summation at each day per sequence and logarithmed if positive. The log of the number of events for a sequence in a day becomes one of the feature inputs to the day after. Furthermore, we define a new feature by aggregating events by date, marking the fraction of daily active users the day prior. This is a dense feature while other features are sparse. If a user was observed for 300 days, it yields 300 time steps of data, even if they were only seen once.

For every dataset, we use 80% of the dates for training and 20% for evaluation. The training target TTE is calculated from the training set keeping training and test temporally separated but sharing sequences. This can be visualized as in Figure 4. We only evaluate on the date immediately after the testing set ends, yielding one prediction per active sequence. This is the same as training a model on currently available data and doing a follow-up study to compare if events occurred as predicted. This differs from prior work as many split into train-test by sequence. Note that temporal models such as CNN and RNN will use past history as input.

## 4.2 BASELINE: BINARY WINDOW MODEL

In order to show the reliability of the model we need to show proper calibration of predicted probabilities and good discriminatory performance. We did this by training individual binary models predicting probability of event within a threshold $\tau = 10, 30, 90, 300$ timesteps ahead. We compared this against HazardNet by querying its predicted distribution on $\theta_t^\tau = \Pr(Y_t < \tau)$. While a binary model is explicitly trained for one threshold, HazardNet needs to learn them all. For this reason one could view the binary models' performance as an upper bound.

While technically not a time to event model, in practice the binary window model is the dominant modeling method for time to event problems. For experimental purposes it is illuminating. No other relevant baseline could be justifiable compared for discrete TTE over different network architectures and different probabilistic queries. The only confounding factor is the treatment of censoring in the training set. HazardNet treats it explicitly while for the binary model there is two choices. One can keep the last $\tau$ timesteps of data (which doesn't have unbiased ground truth about *no event*) or remove it. We used both methods, denoted as Bin and Bin*.

We use standard evaluation metrics for binary classification to evaluate the predicted probabilities. Calibration is evaluated measuring *Expected Calibration error* (ECE) (Naeini et al., 2015), dis-

---

[2]Code for recreating datasets & experiments will be available online

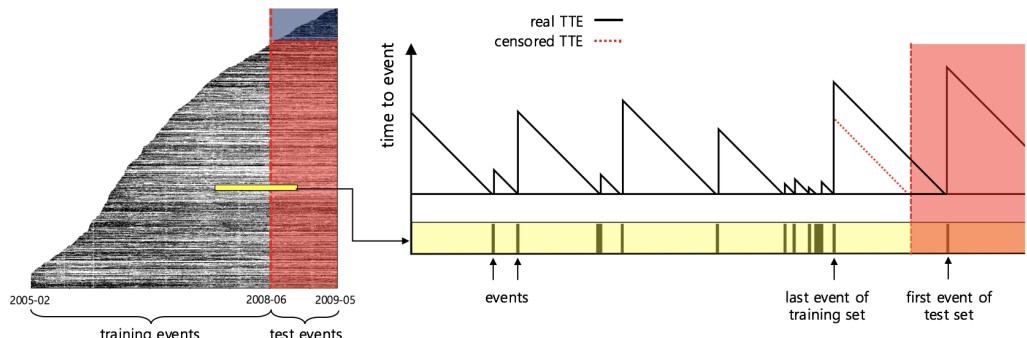

Figure 4: LastFM listening histories as stacked sequences of events sorted by entry into dataset. The training set (a) ends on 2008-06-01, where training TTE is artificially censored as shown as red shades. Test set (b) begins the day after, 2008-06-02. We only report evaluation results comparing prediction made on 2008-06-02 against test set TTE on this day, itself naturally censored by the end of full dataset, 2009-05-09. We don't use sequences shorter than test event lengths, as in blue shades.

Table 3: Highest achieved AUC results on predicting events within 10,30,90 and 300 days after the end of training set with baselines and HazardNet. Bin is the baseline, Bin* is the baseline trained without the last $\tau$ time steps of the training set, Haz for HazardNet. Bold numbers indicate the best result per model and dataset.

| dataset | window | MLP 2x50 | | | RNN 2x50 | | | CNN 14x50 | | |
| --- | --- | --- | --- | --- | --- | --- | --- | --- | --- | --- |
| | | Bin | Bin* | Haz | Bin | Bin* | Haz | Bin | Bin* | Haz |
| lastfm1k | 10 | 0.781 | 0.781 | **0.782** | **0.968** | 0.967 | **0.968** | 0.967 | 0.960 | **0.967** |
| lastfm1k | 30 | **0.759** | **0.759** | **0.759** | 0.966 | **0.967** | 0.965 | 0.957 | 0.955 | **0.965** |
| lastfm1k | 90 | **0.737** | **0.737** | **0.737** | **0.959** | 0.952 | 0.957 | 0.945 | 0.934 | **0.955** |
| lastfm1k | 300 | **0.705** | **0.705** | **0.705** | **0.918** | 0.893 | 0.917 | 0.907 | 0.903 | **0.911** |
| bpi | 10 | 0.687 | 0.687 | **0.690** | 0.798 | 0.809 | **0.845** | 0.799 | 0.807 | **0.834** |
| bpi | 30 | 0.702 | 0.702 | **0.708** | 0.835 | 0.837 | **0.877** | 0.846 | 0.798 | **0.866** |
| linux | 10 | **0.552** | **0.552** | **0.552** | 0.940 | **0.941** | **0.941** | 0.910 | 0.913 | **0.915** |
| linux | 30 | **0.534** | **0.534** | **0.534** | 0.927 | 0.925 | **0.929** | 0.885 | 0.886 | **0.895** |
| linux | 90 | **0.524** | **0.524** | **0.524** | 0.899 | 0.906 | **0.915** | **0.861** | 0.844 | 0.855 |
| linux | 300 | **0.517** | **0.517** | **0.517** | 0.863 | 0.864 | **0.885** | 0.789 | 0.772 | **0.795** |

criminatory performance using *Area Under the Curve* (AUC) and overall performance using *Binary Cross Entropy* (BCE)[3], the loss that the binary model is optimizing for.

We ran an excess of 452 individual training runs. HazardNet was trained repetedly for every architecture and distribution[4] . Binary models were trained for every architecture and threshold, with and without the last $\tau$ timesteps in the training set. More details can be found in Appendix.

### 4.3 RESULTS

HazardNet beats or has identical performance to the binary model for almost every dataset, architecture and distribution. While HazardNet is significantly better on most metrics, the difference is very small. In Table 3 we report the best AUC achieved per experiment. Additional figures and tables can be found in Appendix. We summarize the results here.

**Different datasets have different optimal TTE-distributions:** The 1-parameter exponential distribution was consistently worse than the others. Weibull and WeibullMixedHazards was the least

---

[3]Negative Bernoulli loglikelihood.

[4] 2-parameter Weibull, Pareto, LogLogistic, and their MixedHazard-variants ( 3 mixed CHFs each) and the 1-parameter Exponential distribution.

numerically stable, often failing mid training. LogLogistic seems like a generally good choice. Weibull was optimal for LastFm-1k, differing from other datasets with it's very high event density. Weibull is the only distribution that can model increasing hazard, suitable when lack of events implies event is getting closer. The highly seasonal BPI-dataset was dominated by LogLogistic which can model hazards that peak at some time, possibly suitable to model users certain to return at end of month. The highly unpredictable Linux-dataset, characterized by high dropoff was dominated by Pareto which a strictly decreasing hazard.

**The binary model should not train on the last timesteps:** The temporal CNN and RNN learned that event-probability is artificially low at the end of the training set since we lack negative ground truth samples. This makes the Binary model that were allowed to train on these timesteps poorly calibrated (high ECE and BCE). On the calibration independent metric AUC, the models were indistinguishable except on the seasonal BPI-dataset where we assume the artifactc were more easily overfitted on. This confirms that the binary approach is either biased or misses training data.

**HazardNet beats the baseline overall:** HazardNet was better calibrated (lower minimum ECE) and had better discriminative performance (higher calibration independent AUC). The results are most pronounced for the stronger models (CNN, RNN) implying that our model is most of the time better or equal and less prone to overfit. Comparing the metric the binary model was optimized for (BCE), the results are less convincing but very close. Over all metrics, HazardNet is significantly better, but the difference is very small.

## 5 APPLICATION: MUSICAL ONSET DETECTION

Table 4: F1 scores and standard deviation of 8 folds on onset detection. F1 evaluated before & after predicted probabilities were smoothed using a size 5 hamming window.

| Model | F1 | F1 (Smoothed) |
|---|---|---|
| Schluter & Böck (2014) (our implementation) | 0.853±0.014 | 0.853±0.015 |
| Baseline (Modified Schlüter) | 0.848±0.016 | 0.851±0.017 |
| HazardNet (censored using $c = 10$) | **0.878** ±0.014 | **0.874**±0.015 |
| HazardNet (censored using $c = 20$) | 0.872±0.016 | 0.869±0.018 |

Aiming to show the versatility of the proposed model, we applied it for musical onset detection. The task is to detect relevant musical events in audio signals which is broadly used in applications such as automatic piano transcription (Hawthorne et al., 2017). The state of the art poses it as a binary prediction task. To counter class imbalance and sparseness they apply various smoothing techniques and weights on predicted probabilities (Schluter & Böck, 2014). We propose to soften this problem instead by formulating it as a multivariate time to event problem, jointly predicting the distribution of *time to onset* and *time since onset* (TSE). In this way the onset probability in each timestep can be predicted as in Eq. 5:

$$p_t = \Pr\left(\text{Onset within } \tau = 1 \text{ steps}\right) = \Pr\left([TSE < 1] \cup [TTE < 1]\right) = 1 - S_t^{TSE}(1)S_t^{TTE}(1) \quad (4)$$

Experiments were performed on the Böck dataset, a dataset for evaluating onset detection that is used in several papers (Böck et al., 2012; Schluter & Böck, 2014) The dataset contains 321 audio clips taken from various sources, including piano, violin, percussions, and more (Böck et al., 2012). As feature input we use log scale mel-spectrograms (80 bins) computed with Librosa (McFee et al., 2015) with frequency ranges between 27.Hz to 16kHz. For each audio clip, three mel-spectrograms with different STFT window sizes(23ms, 46ms, 93ms) but same hop size(10ms) were computed and concatenated channel-wise. A single network input is 15 frames of a precomputed mel-spectrogram, where we predict on the center frame. The network architecture is shown in Figure 3 (d). To compare with previous work we used the CNN architecture of Schlüter (Schluter & Böck, 2014). Our model had two extra layers for the parametric output so a comparable baseline model was made to match the number of parameters (Schluter & Böck, 2014).

Target values for training were naturally censored at start and end of song and artificially censored s.t $y < c$ with $y$ the TSE and TTE and $c = 10, 20$. All experiments are trained on 300 epochs, using SGD with a learning rate of 0.001 and momentum linearly increasing at 10 to 20 epochs

from 0.45 to 0.9. To choose predicted onset labels from the predicted probabilities we applied the peak-picking method from Librosa (McFee et al., 2015), using parameters recommended by Böck for offline detection (Böck et al., 2012). For evaluation, we use the onset evaluation metric from mir-eval (Raffel et al., 2014) with 50ms tolerance. We achieved best results using the LogLogistic distribution and $c \leq 10$. Results as average optimal F1-scores from 8-fold cross validation are reported in Table 4. We find that despite only differing by the loss function, our method achieves state of the art results.

## 6 SUMMARY & CONCLUSION

In this paper we presented a simple generalized parametric survival approach for using neural networks to sequentially predict probability distributions over time to the next event.

In the experiments we trained the proposed model (HazardNet) and queried it on classification subtasks for which baseline binary models were explicitly trained for. We found that we often achieve both higher performance and better calibrated probabilistic predictions. Since the model predicts a distribution one can readily calculate other meaningful quantities from it such as predicting quantiles, expected value or sample random TTEs.

While much prior research has focused on specifics of certain probability distributions, we unsurprisingly find that the optimal choice depends on the dataset. We think this emphasizes the importance of a broader discussion on generalized solutions and good abstractions, making it easy to experiment and create new distributions. To this goal, we showed how to train with censored data and how this extends to various continuous, discrete and multivariate distributions with minimal effort.

We introduced and tested a way to compose distributions using their cumulative hazard functions while being able to use censored data (*MixedHazards*-distributions). Despite being much more expressive, we found no proof that they performed better (or worse) than simple distributions. It is tempting to work on trying to encode more information into ever more expressive predicted distributions or fine-grained predicted hazard rates. For the real-life event sequence data we used, we find little to indicate that this is a fruitful path of research.

The main intention of the experiments was to verify that our model makes unbiased, calibrated probabilistic predictions and efficiently utilizes training data. Our criteria for relevant baselines was to be able to train with discrete censored TTE, use asynchronously arriving feature inputs while predicting at each timestep and work with arbitrary neural network-architectures. It should also be able to predict queries such as $\Pr(Y < 10), \ldots, \Pr(Y < 300)$. To the best of our knowledge, no existing work satisfied all of these demands. What was left for fair comparison was the binary window prediction model.

A surprising result was that this baseline makes biased predictions unless one removes the last time-window from the training data. This could have implications for all approaches relying on fixed-window predictions (Harutyunyan et al., 2017; Lee et al., 2018; Luck et al., 2017) who did not discuss or evaluate this apparent drawback.

That aside, comparisons between generalized parametric- and the dominant semi-parametric Cox-Proportional Hazards approach (Katzman et al., 2016; Luck et al., 2017; Joshi & Reeves, 2006) should be priority future work. The latter can only answer a subset of the probabilistic queries but it is theoretically possible to compare models in terms of their predicted rankings between subjects, calculating $\Pr(Y^i < Y^j)$. As we found no prior work on architecture agnostic semi-parametric approaches we propose this as future work.

Finally, for extensions of our work we propose studies on how to extend and evaluate it for asynchronously predicted time to event and how multivariate time to event can be connected to temporal multi-class classification.

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

# A   APPENDIX

## A.1   MORE DETAILS ON EXPERIMENTS OF SECTION 4

To validate our results we made a total of 452 individual training runs. All logs, results and code to recreate the experiments will be released online.

The binary baselines were run for every dataset, threshold ($\tau = 10, 30, 90, 300$) and model (MLP, CNN, RNN) using both the last (biased) binary target value (labeled a 'Bin' in tables) and when removing the $\tau$ last timesteps of the training set (labeled as Bin$^*$). Every setting was repeated for different random seeds; at least 5 times for LastFm1k (98 successful runs in total), 8 times for BPI (64) and at least 4 times for linux dataset (69).

HazardNet was run for every dataset, distribution (Exponential, Weibull, LogLogistic, Pareto and their MixedHazards-versions) and model. Binary evaluation was calculated for every $\tau$. Every setting was repeated for different random seeds; at least 2 times for LastFm1k (63 successful runs in total), 3 times for BPI (76) and at least 3 times for linux dataset (82 runs in total). The TTE was artificially censored s.t TTE $< 300$. This was found to increase numerical stability and reduce overfit. We found little or no advantage in the more expressive often multimodal MixedHazards-distributions. We speculate that this is due to the tasks being inherently hard and exact timings of events unpredictable.

We present summaries of this large comparison matrix below. Here we also show results when using a smaller 1-layer 5-hidden units RNN. Figure 5,6, 7 shows the best metrics (HazardNet vs Binary baseline) evaluated per epoch. It is clear that the HazardNet approach is almost always better or at least as good and seems less prone to overfit. Table 5, 6, 7 summarises the final results in more detail. Figure 8 and 9 shows the test-set loss broken down per distribution. This was evaluated differently (using the whole test set, not just the timesteps after the end of training set). When comparing the binary scores for HazardNet among distributions evaluated in the more rigourous fashion proposed in Section. 4 the overall trends are the same.

Table 5: Binary Cross Entropy evaluated as described in Section 4.

| dataset | threshold | MLP2x50B | MLP2x50B* | MLP2x50H | RNN1x5B | RNN1x5B* | RNN1x5H | RNN2x50B | RNN2x50B* | RNN2x50H | CNN14x50B | CNN14x50B* | CNN14x50H |
|---|---|---|---|---|---|---|---|---|---|---|---|---|---|
| lastfm1k | 10 | 0.462* | 0.462** | 0.461*** | 0.226* | 0.226*** | 0.226** | 0.224** | 0.225* | 0.222*** | 0.228** | 0.236* | 0.226*** |
| lastfm1k | 30 | 0.452* | 0.452** | 0.45*** | 0.222* | 0.221** | 0.219*** | 0.231* | 0.22*** | 0.221** | 0.266* | 0.258** | 0.219*** |
| lastfm1k | 90 | 0.435* | 0.435** | 0.432*** | 0.228** | 0.229* | 0.225*** | 0.252* | 0.249** | 0.224*** | 0.326* | 0.278** | 0.221*** |
| lastfm1k | 300 | 0.36* | 0.333** | 0.329*** | 0.286* | 0.244** | 0.235*** | 0.519* | 0.347*** | 0.264*** | 0.561* | 0.264** | 0.238*** |
| bpi | 10 | 0.627** | 0.628* | 0.625*** | 0.637* | 0.47*** | 0.473** | 1.03* | 0.535* | 0.509*** | 0.867* | 0.558** | 0.508*** |
| bpi | 30 | 0.521* | 0.518** | 0.515*** | 0.808* | 0.355*** | 0.356** | 1.323* | 0.442** | 0.432*** | 1.303* | 0.545** | 0.434*** |
| linux | 10 | 0.179** | 0.18* | 0.179*** | 0.102** | 0.102* | 0.104* | 0.104** | 0.104*** | 0.107* | 0.116** | 0.111** | 0.107*** |
| linux | 30 | 0.294** | 0.295* | 0.292*** | 0.161*** | 0.161** | 0.169* | 0.168*** | 0.169* | 0.18* | 0.216* | 0.188*** | 0.173*** |
| linux | 90 | 0.404** | 0.408* | 0.402*** | 0.22*** | 0.22** | 0.236* | 0.359* | 0.24*** | 0.261** | 0.435* | 0.271** | 0.253*** |
| linux | 300 | 0.527*** | 0.543* | 0.534** | 0.327*** | 0.312*** | 0.351* | 0.643* | 0.362*** | 0.418** | 0.901* | 0.384** | 0.372*** |

Table 6: Area Under the Curve evaluated as described in Section 4.

| dataset | threshold | MLP2x50B | MLP2x50B* | MLP2x50H | RNN1x5B | RNN1x5B* | RNN1x5H | RNN2x50B | RNN2x50B* | RNN2x50H | CNN14x50B | CNN14x50B* | CNN14x50H |
|---|---|---|---|---|---|---|---|---|---|---|---|---|---|
| lastfm1k | 10 | 0.781* | 0.781** | 0.782*** | 0.967* | 0.967*** | 0.967** | 0.968** | 0.967* | 0.968*** | 0.967** | 0.963* | 0.967*** |
| lastfm1k | 30 | 0.759 | 0.759 | 0.759 | 0.965** | 0.965* | 0.966*** | 0.966** | 0.967*** | 0.965* | 0.961** | 0.955* | 0.965*** |
| lastfm1k | 90 | 0.737 | 0.737 | 0.737 | 0.957*** | 0.956* | 0.957*** | 0.959*** | 0.952* | 0.957*** | 0.948** | 0.938* | 0.955*** |
| lastfm1k | 300 | 0.705 | 0.705 | 0.705 | 0.914** | 0.913* | 0.916*** | 0.918*** | 0.893* | 0.917** | 0.907** | 0.903* | 0.911*** |
| bpi | 10 | 0.687** | 0.687* | 0.69*** | 0.837* | 0.852*** | 0.851* | 0.798* | 0.817*** | 0.845*** | 0.799* | 0.807*** | 0.836*** |
| bpi | 30 | 0.702* | 0.702** | 0.708*** | 0.879* | 0.893*** | 0.893** | 0.853** | 0.837* | 0.881*** | 0.847** | 0.803* | 0.866*** |
| linux | 10 | 0.552** | 0.552* | 0.552*** | 0.943*** | 0.943** | 0.942* | 0.94* | 0.941*** | 0.941*** | 0.91* | 0.913*** | 0.915*** |
| linux | 30 | 0.534** | 0.534* | 0.534*** | 0.932** | 0.932** | 0.93* | 0.927** | 0.925* | 0.929*** | 0.885* | 0.887** | 0.895*** |
| linux | 90 | 0.524** | 0.524* | 0.524*** | 0.921** | 0.921*** | 0.919* | 0.899* | 0.906** | 0.915*** | 0.861*** | 0.851* | 0.855** |
| linux | 300 | 0.517** | 0.517* | 0.517*** | 0.897*** | 0.897** | 0.897* | 0.863* | 0.876** | 0.885*** | 0.789* | 0.806*** | 0.795** |

Table 7: Expected Calibration Error evaluated as described in Section 4.

| dataset | threshold | MLP2x50B | MLP2x50B* | MLP2x50H | RNN1x5B | RNN1x5B* | RNN1x5H | RNN2x50B | RNN2x50B* | RNN2x50H | CNN14x50B | CNN14x50B* | CNN14x50H |
|---|---|---|---|---|---|---|---|---|---|---|---|---|---|
| lastfm1k | 10 | 0.003*** | 0.006* | 0.005** | 0.019** | 0.022* | 0.017*** | 0.019** | 0.021* | 0.016*** | 0.017*** | 0.018* | 0.017** |
| lastfm1k | 30 | 0.004*** | 0.009* | 0.005** | 0.019** | 0.02* | 0.017*** | 0.037* | 0.027** | 0.019*** | 0.047* | 0.035** | 0.018*** |
| lastfm1k | 90 | 0.008** | 0.014* | 0.003*** | 0.024* | 0.014** | 0.011*** | 0.056* | 0.044** | 0.019*** | 0.093* | 0.044*** | 0.018*** |
| lastfm1k | 300 | 0.088* | 0.019** | 0.008*** | 0.106* | 0.056** | 0.032*** | 0.18* | 0.097** | 0.056*** | 0.197* | 0.065** | 0.031*** |
| bpi | 10 | 0.029** | 0.034* | 0.021*** | 0.212* | 0.017** | 0.011*** | 0.355* | 0.06** | 0.029*** | 0.304* | 0.062** | 0.019*** |
| bpi | 30 | 0.034* | 0.027** | 0.011*** | 0.368* | 0.016** | 0.008*** | 0.507* | 0.061** | 0.053*** | 0.493* | 0.079** | 0.053*** |
| linux | 10 | 0.012** | 0.013* | 0.012*** | 0.006* | 0.006** | 0.004*** | 0.007* | 0.005* | 0.003*** | 0.013* | 0.005** | 0.002*** |
| linux | 30 | 0.022** | 0.026* | 0.014*** | 0.007*** | 0.007* | 0.008* | 0.015* | 0.009** | 0.007*** | 0.037* | 0.016** | 0.007*** |
| linux | 90 | 0.037** | 0.052* | 0.029*** | 0.007* | 0.004*** | 0.009* | 0.09* | 0.023** | 0.014*** | 0.098* | 0.019** | 0.011*** |
| linux | 300 | 0.028*** | 0.085* | 0.058** | 0.052* | 0.011*** | 0.021** | 0.164* | 0.052** | 0.032*** | 0.195* | 0.015** | 0.012*** |

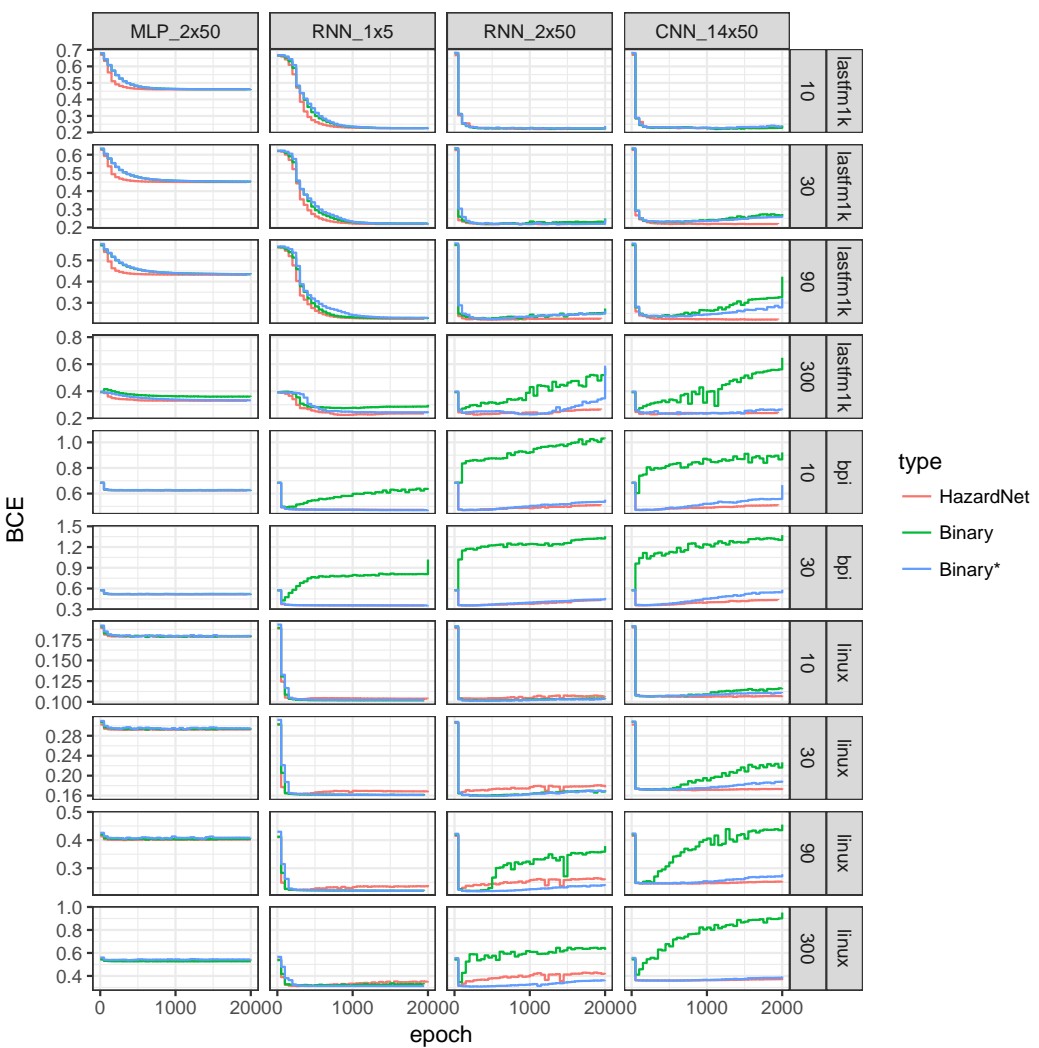

Figure 5: Minimum of each run for the test-loss during training evaluated as described in section 4. Lower is better.

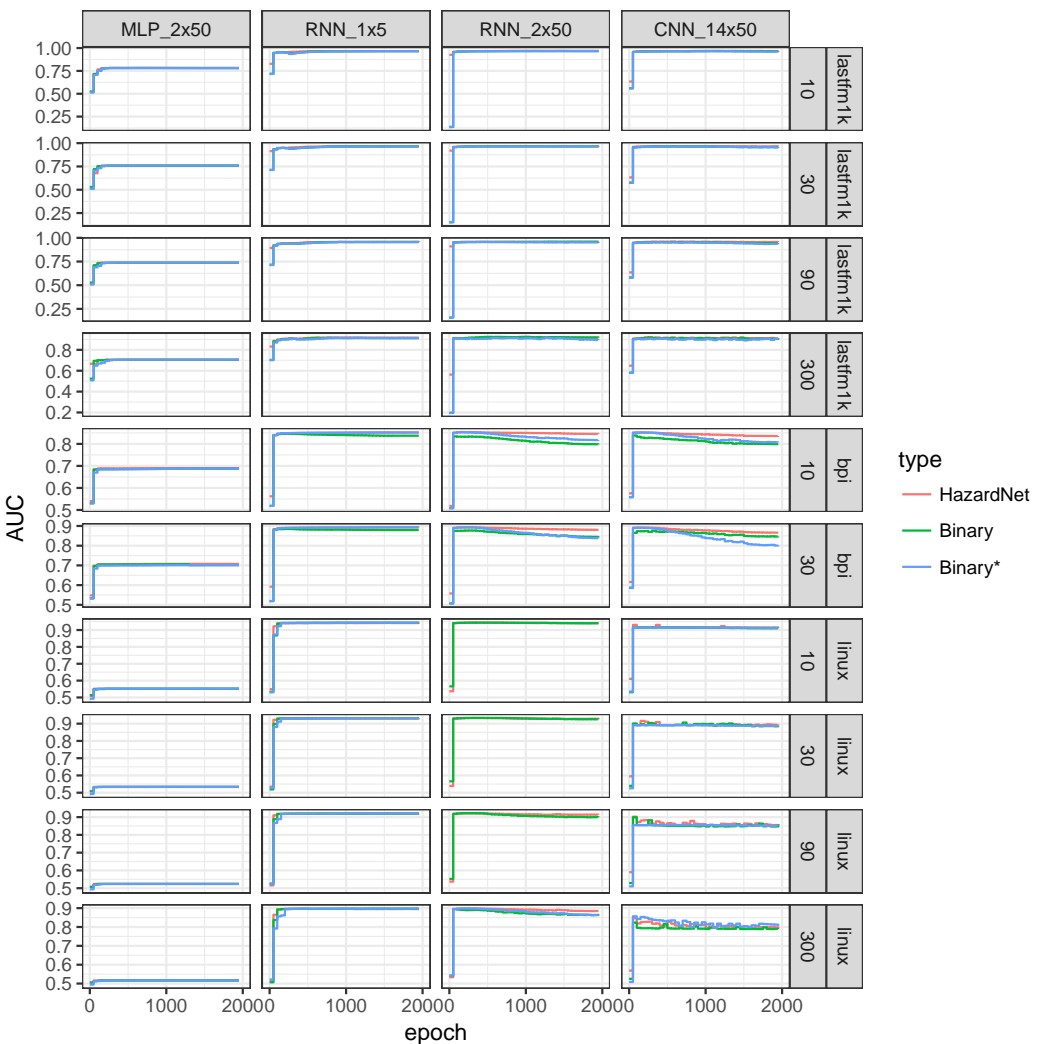

Figure 6: Maximum of each run for the test-loss during training evaluated as described in section 4. Higher is better.

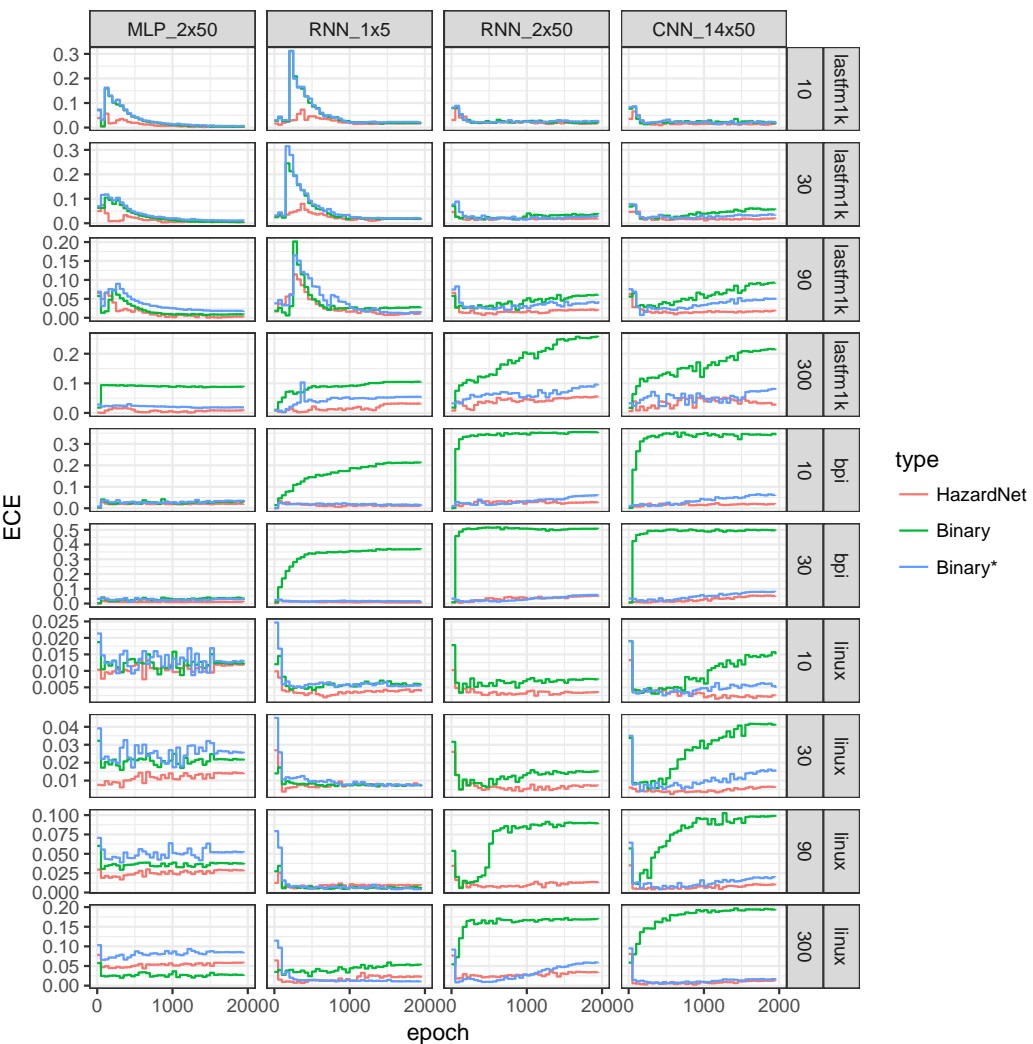

Figure 7: Minimum of each run for the test-loss during training evaluated as described in section 4. Lower is better.

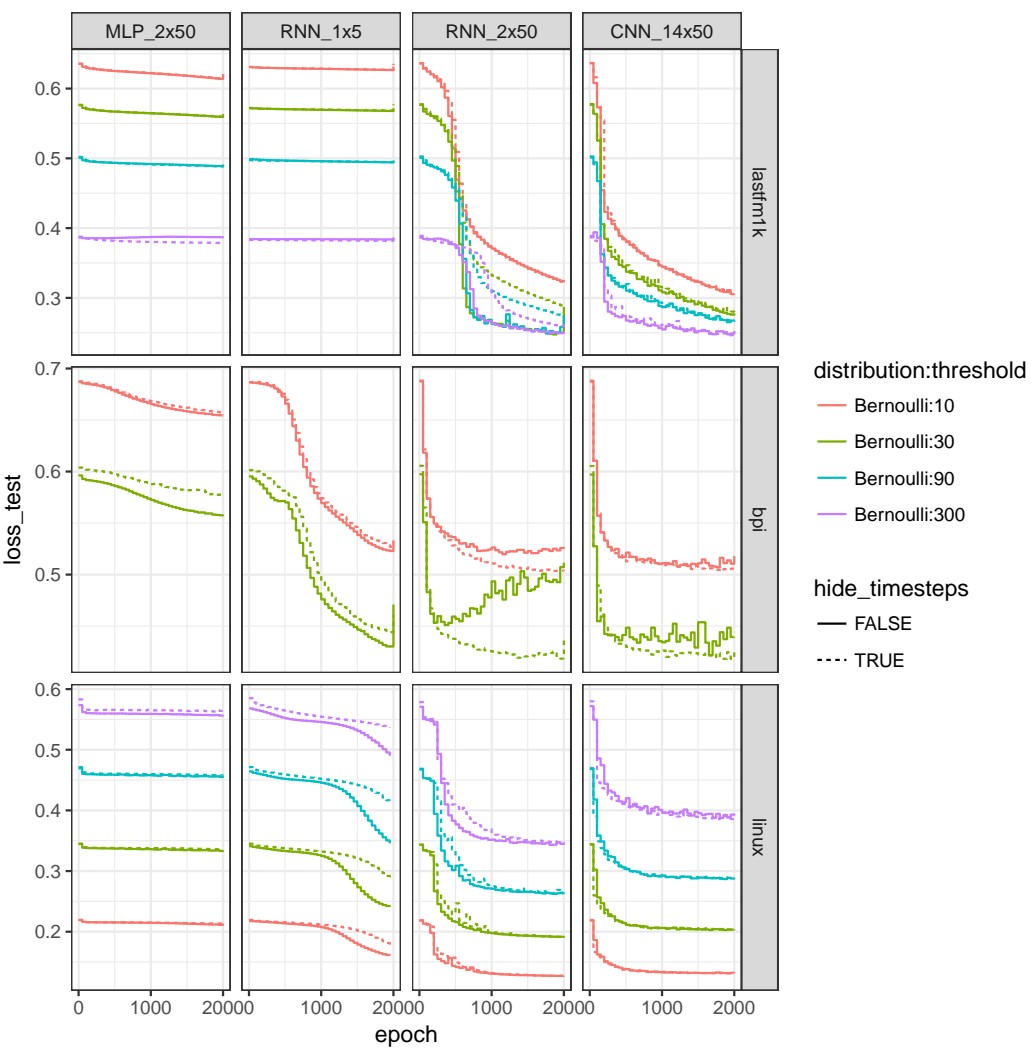

Figure 8: Minimum over each run and timestep for the test-loss during training for the binary baselines (evaluated on all timesteps of the test set)

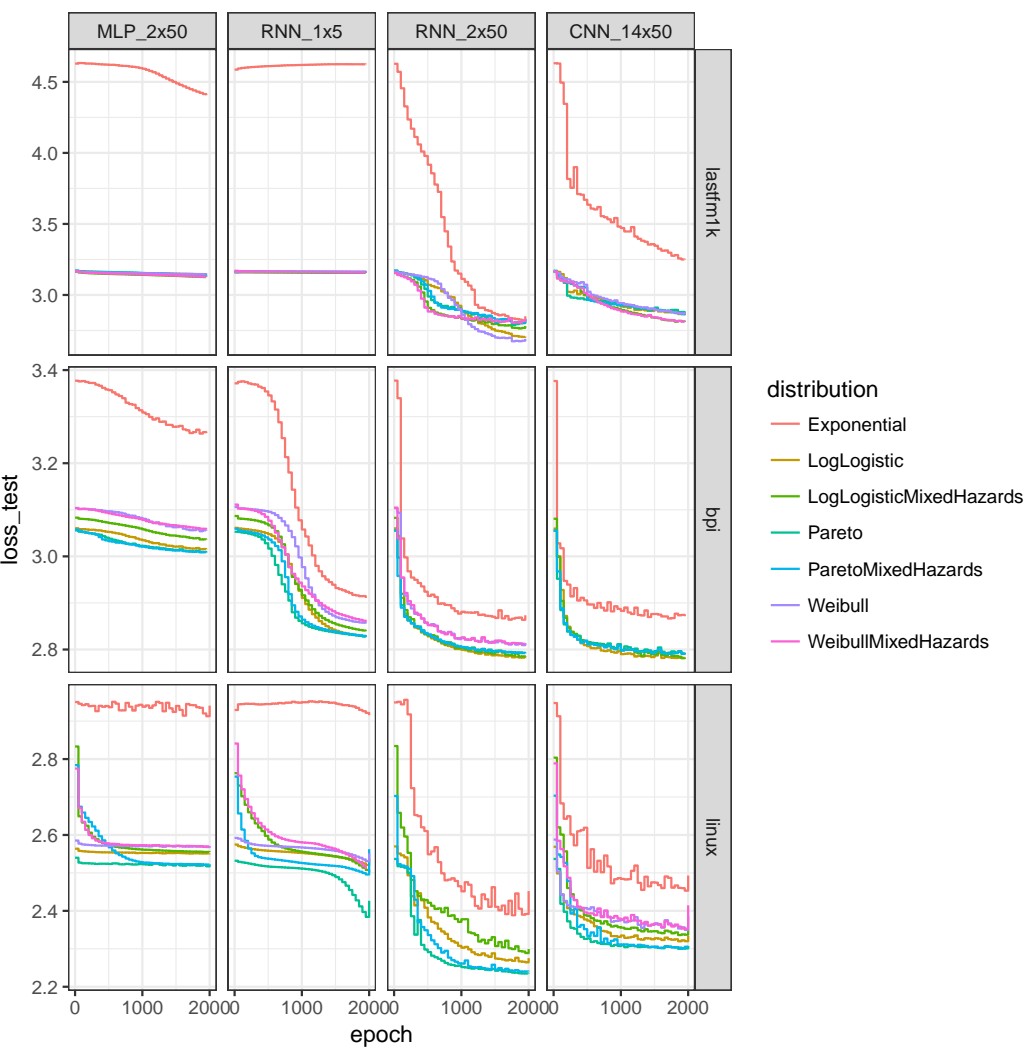

Figure 9: Minimum of each run for the test-loss during training for HazardNet models (evaluated on all timesteps of the test set)

