# OpenReview forum: "Neural Distribution Learning for generalized time-to-event prediction"
_ICLR.cc/2019/Conference_

### Official Review · AnonReviewer3 · 2018-10-24
**Important clarifications should be given about the task and the model**

**Rating:** 3
**Confidence:** 3

**Review:**

The paper "Neural Distribution Learning for generalized time-to-event prediction" proposes HazardNet, a neural network framework for time-to-event prediction with right-censored data.

First of all, this paper should be more clear from the begining of the kind of problems it aim to tackle. The tasks the proposal is able to consider is not easy to realize, at least before the experiments part. The problem should be clearly formalized in the begining of the paper (for instance in the introduction of section 3). It the current form, it is very hard to know what are the inputs, are they sequences of various kinds of events or only one type of event per sequence. It is either not clear to me wether the censoring time is constant or not and wether it is given as input (censoring time looks to be known from section 3.4 but in that case I do not really understand the contribution : does it not correspond to a very classical problem where events from outside of the observation window should be considered during training ? classical EM approaches can be developped for this). The problem of unevenly spaced sequences should also be more formally defined.

Also, while the HazardNet framework looks convenient, by using hazard and survival functions as discusses by the authors, it is not clear to me what are the benefits from recent works in neural temporal point processes which also define a general framework for temporal predictions of events. Approaches such at least like "Modeling the intensity function of point process via recurrent neural networks" should be considered in the experiments, though they do not explicitely model censoring but  with slight adapations should be able to work well of experimental data.

---

> ### Author Response · Authors · 2018-11-27
> **The task is generalized, with two concrete realizations (in experiment).**
>
> > First of all, this paper should be more clear from the begining of the kind of problems it aim to tackle.
>
> We highly value the feedback. An overarching theme was trying to be as general as possible, as we found much work being too specific when there's a much wider set of data, problems, and network architectures that can be utilized once we understand the fundamentals of predicting a parametric survival distribution.
>
> There is however a very common problem domain that we are explicitly working on (Section 4) we also try to generalize the problem to the general task of predicting a probability distribution with possibly censored or discrete target data, and show how it can even improve on common sparse classification tasks (Section 5).
>
> Currently we can’t release implementation and all experimental data without breaking double blind. Once this is done, with corresponding visualizations and data-manipulation should be a bit easier. We tried to make some amends to make it clearer.
>
> > It the current form, it is very hard to know what are the inputs, are they sequences of various kinds of events or only one type of event per sequence. It is either not clear to me wether the censoring time is constant or not and wether it is given as input (censoring time looks to be known from section 3.4.
>
> Censoring time is used to calculate censoring indicators, itself used for training. It typically varies with time. Input to the neural network (features) can be anything, output are parameters of a distribution.
> Specifically, for the experiment in section 4 it's sequences of features. The target (supplied during training) is a sequence of time to event (which will look like a countdown/sawtooth wave as in figure 2) and censoring indicators used for the loss function.
> Censoring time for a sequence would be the time to the end of the sequence (so it's a countdown). Censoring indicators will thus vary.
>
> In section 5, input is a 50 ms time-window of a spectrogram. Target is bivariate; the time to event and time since event (each with their respective censoring indicators). This was supposed to exemplify that with just a change in the output dimension and feature transformation, the same model may be used for something seemingly different like making multivariate predictions.
>
> > but in that case I do not really understand the contribution : does it not correspond to a very classical problem where events from outside of the observation window should be considered during training ? classical EM approaches can be developped for this).
>
> It is true that this corresponds closely to the classical problem of, during training, considering whether events were *not* in the observation/data window (i.e censored). While we heard of no relevant EM-methods but consider our method as exactly developing on a classical approach Parametric Survival Analysis (PSA) which we found other work not sufficiently recognizing.
> The purpose is to say that - while there seems to be many shiny and complex solutions out there - let's first do an in-depth discussion of the classical approach.
> It's easy to see that most other papers can be considered as derivative work of PSA, but we couldn't find anyone going into depth on the idea itself.
>
> Our general reasoning is that by thinking from the classical idea of PSA, many variations (our contributions) immediately follows and are easy to implement  such as predicting all parameters of the distribution like other Density Networks do, being able to discretize TTE, composing distributions to make other distributions, multivariate predictions, architecture agnosticism all the while making it fit well with popular probabilistic programming paradigms (Edward, PyTorch Distributions, Pyro).
>
> > The problem of unevenly spaced sequences should also be more formally defined.
>
> We thought this was clear in the context of event-generated time series no? Asyncronous measurements vs Evenly spaced measurements for temporal models is a classic such problem we wanted to address discussing in section 3.4. In the context of TTE an additional confounding factor is that the lag between observations may be connected to what we want to predict.

---

> > ### Author Response · Authors · 2018-11-27
> > **Why we had only one explicit baseline.**
> >
> > > Also, while the HazardNet framework looks convenient, by using hazard and survival functions as discusses by the authors, it is not clear to me what are the benefits from recent works in neural temporal point processes which also define a general framework for temporal predictions of events. Approaches such at least like "Modeling the intensity function of point process via recurrent neural networks" should be considered in the experiments, though they do not explicitely model censoring but with slight adapations should be able to work well of experimental data.
> >
> > In short, the main differences are not quantitative, and the metrics that can be used for comparing models differ between modeling strategies as the data used and type of predictions made by other methods are limited.
> >
> > Also, the fact that many models would work with censored data using only slight adaptations makes it even more surprising why they seem not to take into account the particular problems that arise when having censored data.
> >
> > Furthermore in the cases when they would work they are are built for particular distributions, neural network architectures, probabilistic queries and types of data. The argument we make is therefore that;
> >
> > 1) Direct comparison is uninteresting.
> > We think that there are enough qualitative differences between models and enough questions about the PSA-Approach that the typical question; "Which model performs best on some metric?" may not be the most interesting one to ask just yet.
> >
> > Our proposition is more that there's many soft qualitative aspects with our approach (simplicity of comparing multitude of neural network architectures to name one) that does not fit into a table.
> >
> > We reasoned that in order to make an interesting and fair comparison we need to factor out maximum amount of confounding factors such as model architecture, which we hard or impossible as most papers are usually about coupling model architecture, data modelling technique and predicted output (Sec. 1-2, 3.4, 4, 6)
> >
> > The question we focused on was more "Is the model unbiased and calibrated?", which we did by comparing it to the binary case. We have not found any convincing work doing this before. We neither find the answers to those questions obvious. Even for the most similar looking work such as https://arxiv.org/abs/1809.02403 they have even come to completely different conclusions, such as the need to weight the censored and the uncensored loss terms.
> > We found that Clipping log-likelihood and properly initializing output layer and making sure the assumptions of PSA holds is sufficient for numerically stable training. See discussion with reviewer #1.
> >
> >
> > 2) Comparison is unfair.
> > Reimplementing our generalized model for most papers using their architecture, predicted distribution, data and queries would often make that implementation mathematically identical to their model (Since they knowingly/unknowingly built upon the fundamental ideas of PSA).
> > Conversely; reimplementing their models for new neural network architectures and making them work for censored discrete data or the type of queries our model supports would often transform them beyond recognition and it would be hard to argue that it still is their model.
> >
> > If we take DeepHit as an example. This proposes an interesting (but arguably complex) method of joining different RNNs together at different time frequencies, each specialized for their own feature type (evenly spaced, asynchronous, event sequences, etc). It predicts multiple types of output in different future timewindows and adds different weighting schemes for the varying loss-terms. They evaluate performance using MAE (which is not defined for censored data), which they base on the paper Recurrent Marked Temporal Point Process (RMTP) (See comments to reviewer #1 about this). There are 10s of examples like this.
> >
> > To compare for example DeepHit's RNN tuned for their data-reshaping and evaluation method (which we critique) on our data with some neural network architecture of our choice one some metric not explicitly optimized for would be misleading at best, and we wonder which question it would answer in the first place.
> >
> > Other methods that have been subject of much recent debate includes temporal point processes such as Hawkes Processes, which has a hand-crafted way of mediating past history to the parameters of the (particular) current distribution of time to event.
> > We wanted to give examples on how to build new processes and distributions, not to verify this on all existing distributions manually.
> > For our model, past events is just one of many possible feature inputs, and as an example we try learning how to map it to the predicted parameters as good as possible using multilayered RNNs or CNNs as an example.
> > Our only claim is that if the current predicted distribution has a Cumulative Hazard Function satisfying certain basic requirements (which a hawkes process clearly does), it works fine.

---

### Official Review · AnonReviewer1 · 2018-10-24
**Deep Hazard Modeling with Mixture of Distributions**

**Rating:** 3
**Confidence:** 4

**Review:**

This paper proposes to use a mixture of distributions for hazard modeling. They use the standard censored loss and binning-based discretization for handling irregularities in the time series.

The evaluation is quite sub-par. Instead of reporting the standard ranking/concordance metrics, the authors report the accuracy of binary classification in certain future timestamps ahead. If we are measuring the classification accuracy, there is a little justification for using survival analysis; we could use just a classification algorithm instead. Moreover, the authors do not compare to the many existing deep hazard model such as Deep Survival [1], DeepSurv [2], DeepHit [3], or many variations based on deep point process modeling. The authors also don’t report the result for non-mixture versions, so we cannot see the true advantages of the proposed mixture modeling.

A major baseline for mixture modeling is always non-parametric modeling. In this case, given that there are existing works on deep Cox hazard modeling, the authors need to show the advantages of their proposed mixture modeling against deep Cox models.

Overall, the methodology in this paper is quite limited and the evaluation is non-standard. Thus, I vote for rejection of the paper.


[1] Ranganath, Rajesh, et al. "Deep Survival Analysis." Machine Learning for Healthcare Conference. 2016.

[2] Katzman, Jared L., et al. "DeepSurv: personalized treatment recommender system using a Cox proportional hazards deep neural network." BMC medical research methodology 18.1 (2018): 24.

[3] Lee, Changhee, et al. "Deephit: A deep learning approach to survival analysis with competing risks." AAAI, 2018.

---

> ### Author Response · Authors · 2018-11-27
> **Binary predictions is a relevant baseline. Mixtures is just a part of our results.**
>
> Thank you for your feedback.
>
> > If we are measuring the classification accuracy, there is a little justification for using survival analysis; we could use just a classification algorithm instead.
>
> A theme in the paper is to point out that predicting a distribution CDF is the same thing as making *all* classification predictions 'Pr(Y<y)' for every timestep 'y>0' ahead.
>
> We think that our experimental results shows (surprisingly) that our survival model outperforms the classification algorithm on the classification task. Both for the arguably contrieved task of predicting specific timesteps ahead (Section 4) or whether a certain timeframe contains an event (Predicting zero steps ahead, section 5). The latter is a well known binary task in its domain which we solve with an arguably novel multivariate survival-formulation.
>
> > The evaluation is quite sub-par. Instead of reporting the standard ranking/concordance metrics, the authors report the accuracy of binary classification in certain future timestamps ahead.
>
> We make a point of our evaluation approach to be non-standard, but we hope that our arguments for it and our critique against the standard evaluation methods for censored sequential problems (Section 3.4, Section 4 and results Section 6) makes sense.
>
> We understand the standard Concordance Index (CI) to estimate how well two predictions are expected to be ordered. A good metric for the dominant paradigm of pointwise-predicting TTE (regression/ranking).
> In contrast, our model predicts a distribution so to answer questions of performance and calibration its arguably not a relevant/helpful metric in its commonly known form.
> To this goal we found the Binary model a good choice of baseline (Section 4), and since CI is not defined for binary predictions, we omitted it.
> As a sidenote, evaluating AUC on different predicted time-windows ahead as we do is tightly related to the time-specific AUC [1][2][3], in turn related to CI.
>
> While possible (but non-standard), we could have generalized CI to compare two predicted distributions directly as 'Pr(Y_i < Y_j)' with ground truth 'y_i<y_j' when available ("concordant"). This however restrics the parametric form of the distributions and is hence less general.
>
> > The authors also don’t report the result for non-mixture versions, so we cannot see the true advantages of the proposed mixture modeling.
>
> This critique was pointed out by other reviewers, and we tried to edit to make this clearer.
>
> The main questions we wanted to answer was not which network architecture or distributions where the best. It was more;
>
> - Does our Parametric Survival model produce calibrated & good predictions independent of choice of architecture and distribution?
> - Is the this approach better or at least as good as explicitly modeling its binary subqueries? (classification approach)
>
> The conclusions was a resounding *yes*.
>
> We tested (but didn't report) all of the following:
>
> (3 datasets) x
> (4 evaluation thresholds ) x
> (3 network architectures) x
> [ Binary x (Use last timesteps or not),
> HazardNet x (4 distributions) x (MixedHazards or not)]
>
> Some per-distribution results can be found in Appendix (see Figure 9) but we could report all tabular statistics broken up by distributions too if interesting.
>
> While not the main question, one conclusion was no significant improvement (see Appendix fig 9) for the more complex multimodal MixedHazards-distributions.
> The reasons can only be speculated about, but it may give hints about the need for predicting fine grained/expressive target distributions, which seems to be quite a concern of current research (Consider DeepHit, Luck et al [0], etc).
>
> While we found this interesting and surprising in its own right, the main benefit we wanted to show was the ease of testing this in the first place using our framework.
>
> > the authors do not compare to the many existing deep hazard model such as Deep Survival [1], DeepSurv [2], DeepHit [3], or many variations based on deep point process modeling.
>
> We do not explicitly test against these, see answer to reviewer #3.
> It should be noted on the other hand, that one of our findings - that binary models will be biased unless removing last timesteps - has consequences for all methods employing what we call "classification" or "multitask" approaches (Ex DeepHit, [0] and more). We tried to clarify this in the revised version.
> Our results implies that unless they preprocessed data as we suggest, their results risks being heavily biased and uncalibrated. If not evaluated as we suggests they won't see this. We can't find any paper commenting on this issue.
>
> [0] https://arxiv.org/abs/1705.10245
> [1] https://www.ncbi.nlm.nih.gov/pmc/articles/PMC5384160/
> [2] https://academic.oup.com/bib/article/16/1/153/238328
> [3] https://www.mayo.edu/research/documents/biostat-80pdf/doc-10027891

---

> > ### Author Response · Authors · 2018-11-27
> > **final note on non-parametrics**
> >
> > > A major baseline for mixture modeling is always non-parametric modeling.
> >
> > This is true, which we agree with (see Section 6 and answer to reviewer #3).
> > We do comment on the qualitative differences (Sec. 2,6) but could not find a reliable method of quantitative comparison.
> > If we would have made a neural-architecture-independent, scalable semi-parametric model working for discrete (read *heavily tied*) sequential data (we found no such work), that would have been the main topic of the paper rather than treating it as a baseline. Instead we propose this as future work and would be very interested to see it.
> > For particular model architectures and data, there's already plenty of such comparisons as you note.
> >
> > In summary, we think that our paper may be an interesting and subtly controversial addition to the discussion on temporal point processes, survival analysis and neural networks and we hope that we've answered your questions.

---

### Official Review · AnonReviewer2 · 2018-11-05
**Re: Neural Distribution Learning for generalized time-to-event prediction**

**Rating:** 4
**Confidence:** 5

**Review:**

The authors propose a parametric framework (HazardNet) for survival analysis with deep learning where they mainly focus on the discrete-time case. The framework allows different popular architectures to learn the representation of the past events with different explicit features. Then, it considers a bunch of parametric families and their mixtures for the distribution of inter-event time. Experiments include a comprehensive comparison between HazardNet and different binary classifiers trained separately at each target time duration.

Overall, the paper is well-written and easy to follow. It seeks to build a strong baseline for the deep survival analysis, which is an hot ongoing research topic recently in literature. However, there are a few weaknesses that should be addressed.

1. In the beginning, the paper motivates the mixtures of distributions from MDN. Because most existing work focuses on the formulation of the intensity function, it is very interesting to approach the problem from the cumulative intensity function instead. Originally, it looks like the paper seeks to formulate a general parametric form based on MDN. However, it is disappointing that in the experiments, it still only considers a few classic parametric distributions. There is lack of solid technical connection between Sec 3.1, 3.2 and Sec 4.

2. The discretization discussion of Sec 3.4 is not clear. Normally, the major motivation for discretization is application-driven, say, in hospital, the doctor regularly triggers the inspection event. However, how to optimally choose a bin-size and how to aggregate the multiple events within each bin is still not clear, which is not sufficiently discussed in the paper. Why is taking the summation of the events in a bin a proper way of aggregation? What if we have highly skewed bins?

3. Although the comparison and experimental setting in Figure 4 is comprehensive, the paper misses a very related work "Deep Recurrent Survival Analysis, https://arxiv.org/abs/1809.02403", which also considers the discrete-time version of survival analysis. Only comparing with the binary classifiers is not quite convincing without referring to other survival analysis work.

4. Finally, the authors state that existing temporal point process work "have little meaning without taking into account censored data". However, if inspecting the loss function of these work closely, we can see there is a survival term exactly the same as the log-cumulative hazard in Equation 3 that handles the censored case.

5. A typo on the bottom of page 3, should be p(t) = F(t + 1) - F(t)

---

> ### Author Response · Authors · 2018-11-27
> **[1-2] Clarifications on Mixtures, note on discretization.**
>
> Thank you for your very relevant and insightful comments.  We appreciate the comments and tried small changes to tie named sections together better. Some things were originally kept orthogonal by design, as distributions (3.1-3.3) and loss functions is quite independent from what kind of feature/target engineering takes places (3.4, 4) which we found previous work being unclear about.
>
> A small comment on the summary;
>
> > The framework allows different popular architectures to learn the representation of the past events with different explicit features.
>
> While this is true, we mainly focus on using features to learn representations of *future* events.
>
> # 1.
> This was also pointed out by other reviewers. We tried to fix this as our presentation was not clear. We did in fact run experiments for ParetoMixedHazards, LogisticMixedhazards, WeibullMixedHazards but did not have space to present results broken down by distributions other than in appendix.
> The was more to show that the CHF-perspective makes it easy to compose positive distributions that effortlessly interfaces with density networks, than to look into details of individual distributions. We tried to make this clearer with some of the edits. See comment to AnonReviewer1 and AnonReviewer2.
>
> In the accompanying github repository (awaiting publication), we will release all logs from each individual experiment. With code released, it should also be clear how this machinery looks like in practice. We did not find a good method to release it yet without breaking a double-blind policy.
>
> # 2.
> We hope not to make a case for any optimal method of choosing the bin width or how to aggregate **features** while discretizing. We think this is an application & data-specific question and want to leave it as such, this is why we wanted to make it easy to make different choices.
>
> In the experiments in Section 4, one of the *features* was 'log(1+count of events in prior timestep)'. While experience shows this is usually a good feature, it should be seen as an arbitrary feature generation step that we chose only because it could be done consistently for all datasets under consideration.
>
> To aggregate/discretize **events**, we consider a timestep with *many events* as a time step with at least one event in. This also naturally fits into the discretization strategy.
>
> If a future timestep (say in 'y' steps) has many events (i.e., is highly skewed), we hope that the current time to event distribution is predicted such that the hazard around that future timestep (equivalently;'Λ(y+1)-Λ(y)') is high.
> In the presented framework, this is left to the modeller as the problem of choosing a reasonable distribution for the task, good feature engineering and the choice of neural networks. We hope our framework makes this easy. Through experiments, we can say that using our approach they should get a calibrated prediction when querying 'Pr(Y<=y)' and that this approach is better or at least as good as modelling it as the binary task of whether events will happen in 'y' timesteps.
>
> There are many design choices that have been found through hard gotten experience that are more style than science. It was hard to motivate them in the paper due to the space limit. We put much effort into harmonizing the notation and the perspective on time to event problems. (Too)Much can be said about this.

---

> > ### Author Response · Authors · 2018-11-27
> > **[4] It's easy to use censored data - but few do it simply or correctly.**
> >
> > # 4.
> > We agree fully with this and think its a strong reason motivating this work. We think that it's completely clear that most loss functions are special cases of parametric survival approach, and modifying the loss in this manner has been known at least since Moivre 1731. Yet we find no work treating it as such or answering basic questions underpinning it or follow what we find to be its implications.
> > On the contrary, there seem to be a widespread belief for the need for applying different weighting schemes, adding extra loss-terms, carefully designing the target value, the network architecture, and similar. If our approach was obvious, then
> >
> > - Why doesn't all work comment on the problem of censored data when it's inherent in the domain?
> > - Why doesn't everybody predict *all* parameters of some distribution (instead predicting only scale-parameter or similar)?
> > - Why are much work distribution-centric and network architectures designed specifically for particular distributions?
> > - Why is discrete data often treated as a special case?
> > - Why does the presentation/notation of the loss functions so widely differ?
> > - Why is it that there are many papers who uses Neural Networks (NNs) to explicitly model the hazard function but doesn't compare to parametric density network baselines (as the one we propose) with similar NN-architectures/number of parameters?
> > - Why are stochastic process (Temporal Point Process)-based models with restricted ways of taking account of the nonlinearities of feature data still considered relevant baselines to advanced NNs?
> > - Why are evaluations mainly limited to pointwise predictions?
> > - Why are pointwise-predictions evaluated with metrics that does not work for censored data (Ex MAE as for DeepHit)?
> > - Why is there so little discussion regarding the calibration of predicted distributions?
> > - Finally, why is a model with strictly increasing hazard function (RMTP [1]; λ(x) ~= exp(x/L) still considered among SOTA? No hazard function that we trained which can take that form (ex Weibull) seem to want take it for similar data. Hint: It's about evaluation. If one removes censored data one cuts the right tail of the empirical distribution so hazard is *by design* increasing in the training data since each sequence now ends with an event, so the RMTP hazard function fits by design.)
> >
> > The point is not to critique prior work, clearly the concepts of parametric survival analysis is widely known but maybe its general implications for how it effortlessly fits with density networks or discrete data is less known? It might have to do with Time To Event modelling being inherently complex and confusing.
> >
> > In relation to whether we should have added comparisons to other work we refer to the answer we gave to reviewer #3. In short, we would have wanted to compare explicitly to other work, but this was found hard or irrelevant to fit in. Instead we limited ourselves to digging deep into whether this general (Parametric Survival Approach) is correct by considering if it's performant (vs Binary classification) and calibrated.
> >
> > [1] https://www.kdd.org/kdd2016/papers/files/rpp1081-duA.pdf

---

> > > ### Author Response · Authors · 2018-11-27
> > > **[3] Example: Seemingly similar PSA-work with provable misconception.**
> > >
> > > # 3.
> > > We added this to Section 2. We thought it had many qualities, especially that it seems to be the first work comparing something very close to a (discrete) parametric survival approach with other approaches using what we understand as comparable NN-architectures for each loss function.
> > >
> > > It was overlooked at first because this is one of many papers using weighting schemes for the loss function without evaluating the unbiasedness (read; calibration) of the approach.
> > > In short, the paper (as many others) suggests weighting the loss with '0<a<1' as:
> > >
> > > 'loss = a*uncensored_part+(1-a)*censored_part'
> > >
> > > One can show analytically that the weighting scheme will lead to biased predictions and we didn't want to comment on it prematurely as it was just published, but let’s make this point here.
> > >
> > > As a proof, consider the case of the exponential distribution, with 'L' being the scale-parameter, 'Y~exp(L_real)', and 'a' being the proposed loss-weighting scheme, then the expected value of the gradient of the log-likelihood with censoring;
> > >
> > > '''
> > > d/dL E[-a*<Y<c>* log[L]-u(1-a)min(Y,c)/L] =
> > >   F(c)[(1-a)-a*(L_real/L)]/L = *
> > > '''
> > > Where '<Y<c>' is the iverson bracket, being '1' for an uncensored sample.
> > >
> > > In other terms, with 'L_real=L' then '*=(1-exp(-c/L))[1-2a]/L' which needs to be '0', so it will converge to 'L_real=L' iff 'a=0.5'. Otherwise it converges to 'L = L_real*(1-a)/a'
> > >
> > > If the math is not convincing, let's apply that weighting scheme just for parametric learning in a simple experiment:
> > >
> > > '''
> > > import torch
> > > import torch.nn as nn
> > >
> > > def train_censored(a=0.5,c=1):
> > >   torch.manual_seed(1)
> > >   # True distribution is ~Exp(1)
> > >   L_real = torch.ones(1)
> > >   dist = torch.distributions.Exponential(rate=L_real)
> > >   L_log = nn.Parameter(torch.randn(1))
> > >   optimizer = torch.optim.SGD([L_log,],lr = 1)
> > >   for step in range(3000):
> > >     y = dist.sample((1000,))
> > >
> > >     # Censored (Truncated) as [0,c)
> > >     y = y = torch.min(y,y*0+c)
> > >     u = (y<c).float() # "non-censoring indicator"
> > >     optimizer.zero_grad()
> > >
> > >     L = L_log.exp() #
> > >     # Exponential Log-likelihood with "weighted loss terms"
> > >     loglik = -u*a*L.log()-(1-a)*y/L
> > >     loss = -loglik.mean()
> > >
> > >     loss.backward()
> > >     optimizer.step()
> > >   print('Sought=',L_real.item(),'\tResult:\t',L.item(),'\tExpected=',(L_real*((1-a)/a)).item())
> > >
> > > train_censored(a=0.5,c=3)
> > > #>> Sought= 1.0 Result: 1.0122132301330566 Expected= 1.0
> > > train_censored(a=0.9,c=3)
> > > #>> Sought= 1.0 Result: 0.11424004286527634 Expected= 0.1111111119389534
> > > train_censored(a=0.1,c=3)
> > > #>> Sought= 1.0 Result: 8.986083030700684  Expected= 9.0
> > > '''
> > >
> > > I.e only with equal (no) weighting will it converge to a correct estimate lambda = 1.
> > >
> > > We think this example illustrates that there still seems to be some confusion about whether the basic approach we argue for (not weighting or manipulating the loss function) works in the first place, and whether it’s in fact the correct way of doing it. Considering that many of the apparent applications of these types of models are clinical, this is insufficient.
> > >
> > > The point here is that maybe the fundamentals of survival analysis may not be as widely understood as we would like to think it is, and that there's still space to ask questions beyond comparing full implementations of models and rank them on some metric. This is what we wanted to achieve with our paper.
> > >
> > > #5.
> > > Thank you, we corrected this.

---

### Meta-Review · Area_Chair1 · 2018-12-14
**Meta-Review for Neural Distribution Learning**

**Confidence:** 5
**Recommendation:** Reject

**Metareview:**

All reviewers agree to reject. While there were many positive points to this work, reviewers believed that it was not yet ready for acceptance.